# Cultural Adaptation and Psychometric Properties of the Diabetes Quality of Life Scale in Afaan Oromoo among People Living with Type 2 Diabetes in Ethiopia

**DOI:** 10.3390/ijerph18147435

**Published:** 2021-07-12

**Authors:** Dereje Chala Diriba, Doris Y. P. Leung, Lorna K. P. Suen

**Affiliations:** 1School of Nursing, Hong Kong Polytechnic University, Kowloon, Hong Kong; 19048615r@connect.polyu.hk (D.C.D.); doris.yp.leung@polyu.edu.hk (D.Y.P.L.); 2School of Nursing, Tung Wah College, Kowloon, Hong Kong

**Keywords:** diabetes quality of life, DQOL, Afaan Oromoo, psychometric properties

## Abstract

Background: The original 46-item diabetes quality of life (DQOL) scale has been translated into different languages, and the translated DQOL has shown good reliability and validity after deleting some items. The aim of this study was to translate the diabetes quality of life (DQOL) scale into Afaan Oromoo and to culturally adapt and evaluate the psychometric properties of the DQOL-Afaan Oromoo (DQOL-AO) among people living with T2D in Ethiopia. Methods: A cross-sectional study with a convenience sampling technique was conducted in 2020. The DQOL was translated and adapted to Afaan Oromoo. Item–total correlations and exploratory factor analysis (EFA) assessed factor structure; the Cronbach’s alpha assessed internal consistency and relationships with gender, educational status, marital status, age, and employment status; and status of diabetes-related disease assessed the construct validity of the DQOL-AO. Results: 417 participants responded to all items of the DQOL. Item–total correlation analysis and EFA produced a 34-item DQOL-AO with four subscales, which demonstrated that the internal consistency of the overall DQOL-AO was 0.867, and scores were 0.827, 0.846, 0.654, and 0.727 for the impact, satisfaction, social/vocational worry, and diabetes-related worry subscales, respectively. Statistically significant differences between QOL were obtained in educational status (F = 7.164, *p* < 0.001) and employment status (F = 4.21, *p* = 0.002). Individuals who attended college and above and government employees had better QOL. Conclusion: The 34-item DQOL-AO provided preliminary evidence as a reliable and valid tool to measure diabetic-related QOL before it can be widely used among adults living with T2D who speak Afaan Oromoo.

## 1. Introduction

Diabetes is one of the most prevalent non-communicable diseases and the cause of a major global public health problem [1]. The International Diabetes Federation (IDF) estimates that 463 million adults worldwide have diabetes; four million of them die every year. Diabetes is widespread globally, with the IDF projecting that there will be 700 million adults living with diabetes in 2045. The burden of the disease is devastating in middle- and low-income countries, with a prevalence of 13.5%. An alarming 143% increase is expected in Africa, which will have 47 million cases—the highest predicted increase of all the IDF regions—if crucial action is not taken immediately. Over 19 million adults were living with diabetes in Africa in 2019, with Ethiopia accounting for 1.7 million (3.2%) [2].

Quality of life (QOL) in people living with diabetes becomes a daily goal and is considered an important treatment outcome [3]. QOL assesses the physiological well-being, physical, and psychosocial aspects, and lived experience of patients. Several studies have reported that diabetes negatively affects a person’s QOL [4,5,6]. Diabetes poses social, physical, sexual, and physiological impacts, and these impacts are worse if complications of diabetes develop [5,7]. Although QOL is recognized as an important patient-reported outcome, it is rarely assessed in diabetes research [8]. A recent study of 25 years’ experience of the impact of diabetes assessment study pointed out that the QOL needs to be addressed by researchers as a priority [9].

Measuring QOL in people living with diabetes is imperative [4]. Even though various forms of diabetes-specific QOL measures are available [10,11,12,13], the diabetes quality of life (DQOL) scale provides a comprehensive assessment of the components of QOL among people living with diabetes in general; it has been widely used in different studies to measure QOL among people living with diabetes [14,15,16,17] and is sensitive to disease severity and lifestyle changes [18].

The DQOL scale was originally developed in 1988 by the diabetes control and complications trial research group in English, in a study aimed to evaluate the effects of two different diabetes treatment regimens on QOL. The DQOL has 46 items measuring four domains: satisfaction, impact, social/vocational worry, and diabetes-related worry, with a total Cronbach’s alpha of 0.92 [19]. Subsequently, numerous shortened versions of the DQOL were produced in different languages [14,17,20,21,22]. A 24-item Chinese version [17], a 44-item Brazilian DQOL version [21], an eight-item Brazilian brief version [20], a 45-item Turkish version [22], a 46-item Iranian version [23], and a 13-item Malay revised version were developed [14]; all showed acceptable validity and reliability. Consistent with the original version of the DQOL, all the shortened versions of the DQOL have four domains, with one exception: the 13-item Malay revised version [14] measures three domains, namely satisfaction, impact, and worry. Some studies have also reported that some items of the DQOL had a low correlation with items in the same domain [15,17], thus it is deemed necessary to establish the correlation matrix among items in the domain. The DQOL scale Cronbach’s alpha of these versions ranged from 0.702 to 0.92 [19,20,22,23].

The construct validity of the various DQOL studies demonstrated that educational status, employment status, age, and comorbidity status were significantly related to QOL among adults living with diabetes. Specifically, people living with diabetes who were female and married scored significantly higher than their male counterparts, while people living with diabetes who had not attended formal education, were older, were separated/widowed, were unemployed, and people living with diabetes complications scored significantly lower in QOL [18,24,25,26].

Though the DQOL scale is available in different language versions, there is no translated, culturally adapted, and psychometrically validated version in Afaan Oromoo, which is the most widely spoken language in Ethiopia (33.8%) and is the fourth most widely spoken language in Africa [27]. Hence, this study aimed to translate the original English version of the DQOL into Afaan Oromoo and culturally adapt and evaluate its factor structure, reliability, and construct validity among adults living with Type 2 diabetes (T2D) in Ethiopia.

## 2. Materials and Methods

### 2.1. Study Design

A cross-sectional study design was employed to examine the psychometric properties of the Afaan Oromoo version of the DQOL measure.

### 2.2. Participants

A convenience sampling technique was used to recruit people living with T2D attending the diabetes center of Nekemte Specialized Hospital in Western Ethiopia for their monthly medical check-up between June and August 2020. Included among the people living with T2D were those who (1) had been treated at the diabetes center in the hospital for six months or more; (2) were in a stable medical condition; (3) were aged 18 or over; (4) were cognitively intact, and (5) were able to speak and understand Afaan Oromoo. People living with Type 2 diabetes were excluded if they had a hearing problem.

Table 1 presents the sociodemographic characteristics of the participants. The response rate of the participants was 90.6% (417/460) and cases with incomplete data were excluded from analysis. The mean age of the participants was 50.2 years (SD ± 11.7); 51.3% were female, 77.5% were married, and 88.2% were Oromoo. More than half (56.8%) were Protestant Christian, and the majority (61.6%) received support from their spouse. One-third (33.1%) of the participants had attended ≤grade 8 and 27.2% were employees of a private organization. More than half (55.4%) of the people living with diabetes had comorbid diseases, and nearly half (45.6%) had hypertension. More than three-quarters (82.7%) had ≤10 years of history of diabetes.

Ethical approval for the study was obtained from The Hong Kong Polytechnic University. Permission to collect data was obtained from the hospital administrator before the start of the study and informed written consent was obtained from the participants. The confidentiality of the data was ensured through coding.

### 2.3. Translation of the DQOL

Permission to translate and adapt the 46-item DQOL scale was obtained from the scale developers. The DQOL was translated and culturally adapted into Afaan Oromoo according to the six-stage recommendation of cross-cultural adaptation developed by the Institute for Work and Health in 2007 [28]. In stage 1, two versions of the forward translation of the original version of the DQOL were prepared by two translators (a health professional and a naïve translator who is a Ph.D. holder in Afaan Oromoo). In stage 2, a synthesis of the translations obtained in stage 1 was made by the principal researcher, and a reconciled translation of the scale was developed after agreement on any discrepancies was reached. In stage 3, two separate versions of the back-translation of the scale were prepared by another two excellent translators, who were Ph.D. holders in English and native speakers of Afaan Oromoo. In stage 4, an expert panel consisting of seven professionals (one public health expert, one nurse, one Afaan Oromoo language expert, two forward, and two backward translators) was formed and they evaluated conceptual, semantic, and idiomatic equivalences of the translated versions of the scale using five-point Likert scale items to calculate the content validity index (CVI). Discrepancies were resolved through discussion until consensus was reached. The CVI of the Afaan Oromoo version of the DQOL tool was ≥0.95. In stage 5, 30 people living with T2D were asked to assess the applicability, readability, and clarity of the item content of the expert-evaluated version of the scale [29]. The cultural adaptation was made using locally spoken and acceptable words. The people living with diabetes were requested to suggest the appropriate terms, and amendments were done to the local culture. In stage 6, an amendment was made based on feedback from the participants, using appropriate words and restructuring some sentences in a culturally appropriate way, and the final version of the scale in Afaan Oromoo, the Diabetes Quality of Life—Afaan Oromoo (DQOL-AO), was developed and subjected to psychometric testing.

### 2.4. Sample Size Calculation

For psychometric testing, the required sample size was determined based on exploratory factor analysis (EFA) using a widely used case-to-variable (rule of thumb) ratio of 10:1 [30,31]. EFA is used because many previous validation studies have reported variations in the factor structure of the scale in populations with different languages [17,32], hence it is better to use EFA to explore the factor structure of the scale in the current target population, which uses a different language. Hence, the minimum required sample size was 460.

### 2.5. Instrument

The 46-item DQOL has four major domains: satisfaction (15 items), impact (20 items), social/vocational worry (7 items), and diabetes-related worry (4 items). Items in the satisfaction domain are scored on a five-point scale ranging from 1 (very satisfied) to 5 (very dissatisfied), and items in the impact and the two worry domains are scored on a five-point scale, ranging from 1 (no impact and never worried) to 5 (always impacted and always worried). If an item is not relevant to the respondent, the ‘Does not apply’ option is provided for the social/vocational worry and diabetes-related worry subscales and will not be scored. A lower score in DQOL indicates a better QOL [18].

Sociodemographic variables, namely gender, marital status, ethnicity, religion, educational level, a family member usually providing support, and employment status, and patient-related factors such as the diabetes-specific complication (s) and year of first disease diagnosis were collected.

### 2.6. Data Collection Procedure

Eight data collectors who have experience in data collection were trained in a one-day workshop to ensure they were familiar with and understood the items in the scale and the techniques of conducting interviews for the study. People living with diabetes were approached when they were waiting to see the doctor in the diabetes center of the hospital. After explaining the purpose and study procedure, the data collectors screened the people living with diabetes for their eligibility. Having obtained their informed written consent, the data collectors then administered the questionnaire via face-to-face interviews.

### 2.7. Statistical Analysis

All data analyses were conducted using SPSS statistics version 25. The factor structure of the DQOL-AO was examined in two steps. In step 1, item reduction based on the item–total correlation was performed. Any item with an item–total correlation coefficient below 0.3 was removed [31,33]. In step 2, EFAs were conducted on the items remaining after step 1. The Kaiser–Meyer–Olkin (KMO) and Barlett’s tests checked for the appropriateness of conducting EFA. The factor retention was based on four criteria: (i) eigenvalues > 1; (ii) scree plot; (iii) interpretability of the retained factors; and (iv) factor loadings > 0.4. For items cross-loaded on factors, the retention of the item to the factor was determined by two criteria: (1) a higher loading effect of the item onto the factor and (2) the interpretability of the item. The Cronbach’s alpha value was then calculated to assess the reliability of the subscales and the overall scale of the DQOL-AO. Ceiling and floor effect analysis was performed for the subscales and the overall scale to distinguish the proportion of respondents with the highest and lowest QOL scores, respectively [34]. Ceiling or floor effects were judged if more than 15% of subjects reached the highest or lowest score, respectively [35].

The construct validity of the DQOL-AO was assessed by the known group and correlation analysis—Pearson’s correlation was used for continuous demographic variables and an independent t-test or ANOVA was used for categorical variables. In all the analyses, a *p*-value < 0.05 was considered statistically significant.

## 3. Results

The scale was translated and culturally adapted according to the cross-cultural adaptation guidelines stated under Section 2.3. Disparities were reported among the experts on the wording of some items and the correctness of others. For example, one expert commented with respect to Item 10, regarding the impact on the individual’s sex life, that “complications related to type 2 DM, like impotence, is something that is gradual and permanent. It is not something that comes and goes”. It was explained that the ‘always’ response is an option. Thus, if the individual has sexual-related problem, he/she may respond ‘always’. Varying opinions on other items were solved through discussion among the translators and subject experts.

### 3.1. Factor Structure

#### Item–Total Correlation

Table 2 shows the item–total correlation statistics of all items, 12 of which had item–total correlations <0.3, indicated in bold. These included two items (items 7 and 15) from the *satisfaction* subscale; seven items (items 3, 8, 16, 17, 18, 19, and 20) from the *impact* subscale; two items (items 4 and 5) from the *social/vocational worry* subscale; and one item (item 4) from the *diabetes-related worry* subscale. Satisfaction item 7, which talks about satisfaction with the knowledge of diabetes, might not be removed due to their belief that they know about diabetes. Impact items 3, 8, and 17 were removed in this version, possibly because this population knows about low glucose levels, self-esteem, and respect. Similarly, impact item 20 was removed, possibly since the vast majority of people living with T2D do not receive insulin, hence an insulin reaction is not expected and is not a relevant item [36]. The items were removed from the AO version of the DQOL. A total of 34 items were retained in the scale: satisfaction (13 items), impact (13 items), social/vocational worry (6 items), and diabetes-related worry (3 items). These items were subjected to EFA.

### 3.2. Exploratory Factor Analysis

An EFA was conducted to examine the factor structure of the remaining 34 items of the DQOL-AO. The KMO statistic was 0.865 and Bartlett’s test statistic was 5739.562 (*p* < 0.001), implying sampling adequacy and appropriateness for factor analysis. The scree plot and eigenvalues suggested four possible factor solutions, namely, 4-, 5-, 6-, and 7-solution. Based on the interpretability of the factors, the 4-factor solution was selected because it produced four explicit factors that resembled the original DQOL. The findings of EFA showed a 4-factor solution comprising 45.12% of the total variance retained. All the factor loadings of the 4-factor solution were greater than 0.4, hence all 34 items in the DQOL-AO were retained.

The items were examined with their factor loadings, and any item that loaded on (an)other subscale(s) in addition to its original subscale was retained in its original subscale only to enhance the interpretability. As indicated in Table 3, 13 items were retained under the impact subscale, 13 in the satisfaction subscale, five in the social/vocational worry subscale, and three in the diabetes-related worry subscale. All 34 items were retained as the DQOL-AO version by EFA.

The items that were retained in the DQOL-AO version (in Afaan Oromoo) are presented in Table 4, and individuals interested in the tool can use it for clinical evaluation and research purposes with appropriate acknowledgement of the source.

### 3.3. Reliability Estimate

Table 5 below presents the internal consistency, ceiling, and floor effects of the DQOL-AO. The finding of the DQOL-AO shown good internal consistency (>0.7) in three subscales (impact, satisfaction, and diabetes-related worry), but the social/vocation worry subscale has questionable internal consistency (α = 0.654).

The ceiling and floor effects of the items were calculated for the scale and subscales of the DQOL-AO version. A very small proportion of the people living with diabetes (≤1.0%) attained the highest QOL score, and ≤6.2% of the study participants achieved the lowest QOL score in all four subscales. For the overall DQOL-AO scale, there was also no evidence of any ceiling or floor effect (0.2%).

### 3.4. Construct Validity

Table 6 indicates the construct validity results and direction, with bold figures indicating better quality of life. The results of ANOVA showed that education status (F = 7.164, *p* < 0.001) and employment status (F = 4.211, *p* = 0.02) demonstrated a significant difference in the DQOL-AO scores of the participants. There was a significant difference between participants who attended college and above (2.23 ± 0.40) and those who had not attended formal education (2.43 ± 0.43, *p* = 0.008) or attended only elementary school (2.47 ± 0.42, *p* < 0.001). However, those who attended college and above had better QOL. There was also a significant difference between government employees (2.23 ± 0.39) and those who were retired/disabled (2.44 ± 0.39, *p* = 0.014) or farmers (2.39 ± 0.44, *p* = 0.002): better QOL was revealed among government employees. There was no statistically significant difference in QoL with other variables.

By contrast, diabetes-related complication(s) (*t* = −1.397, *p* = 0.163), marital status (F = 1.047, *p* = 0.352), and gender (*t* = −1.064, *p* = 0.288) showed non-significant results. In addition, a non-significant but positive correlation between age and DQOL-AO (r = 0.057, *p* = 0.242) was obtained.

## 4. Discussion

The original version of the DQOL scale was translated to Afaan Oromoo according to cross-cultural adaptation guidelines. The content validity of the translated version was evaluated by experts and demonstrated acceptable CVI [37]. The DQOL-AO version was found to be reliable and valid to measure QOL among adults living with T2D who speak Afaan Oromoo. The DQOL-AO consists of 34 items, and a four-factor solution was retained in the EFA. The four factors found were consistent with the original, Brazilian, Brazilian brief, Chinese, and Turkish versions of the scale [17,19,20,21,22], but inconsistent with the Malay version [14,15].

As indicated in Table 3, six items from satisfaction, six items from impact, two items from social/vocational worry and one item from the diabetes-related worry scale were removed from the original scale. Two main reasons may account for dropping these items. First, some of the deleted items may be of less important for assessing the QOL of adults living with diabetes. The seven dropped items (impact items 3, 16, 19, and 20; social/vocational worry items 4 and 5; and diabetes-related worry item 4) in this DQOL-AO version were consistent with the deleted items in the revised Malay version [14]. On the other hand, the removed satisfaction item 7; impact items 8, 16, 17, 19, and 20; and social/vocational worry item 5 accord with the items removed in the Chinese version [17]. Four items: satisfaction item 7, and impact items 3, 8, and 17, were removed in this version, also consistent with items removed in the Chinese and brief DQOL-Brazil-8 versions [17,20]. Second, these items are redundant in terms of expressing QOL among adults living with diabetes. For example, the satisfaction items 7 and 15 and impact items 4 and 16 which is similar with Malay version [14]. The social/vocational worry items 4 and 5 were removed from the subscale. This might be because most respondents were employed or retired and might not worry about employment. The dropping of these items implies that they are less important to measure the satisfaction and impact parameters among people living with diabetes who understand Afaan Oromoo. The DQOL-AO is a short scale developed with good psychometric properties; hence it is potentially appropriate for assessing QOL among Oromoo people living with type 2 diabetes, especially in busy clinical settings.

The DQOL-AO also had good internal consistency. This was similar to the Turkish and Chinese versions [17,22]. The impact subscale showed better internal consistency when compared with the original [19] and a revised Malay version [14], and was consistent with the Chinese [17] and Turkish versions [22]. The satisfaction subscale’s internal consistency revealed good reliability that was in line with the original version [19] and inconsistent with the revised 13-item Malay, Malay, Chinese, and Turkish versions [14,15,17,22]. While diabetes-related worry demonstrated good internal consistency, that of social/vocational worry was questionable. These findings were inconsistent with the Chinese, Turkish, Malay, and revised Malay versions [14,15,17,22], but the low level of internal consistency resembled the original version [19]. The possible variation in internal consistency in worry may be because most people living with diabetes were married and employed. Another explanation for the low level of internal consistency among worry items might be that there were fewer items in the domain [38]. Additionally, this scale addresses the more specific concern of the patient’s perceptions of diabetes-related psychological distress [19].

As shown in Table 6, the known-group analyses showed that educational and employment status were significant predictors of QOL among people living with T2D. Attending college and above education and working in government institutions were related to better QOL. These findings were consistent with the study conducted in Botswana and Gondar, Ethiopia among people living with diabetes, which revealed that those who were educated and employed had better QOL [26,39]. No significant correlation was established between the age of the patient and overall QOL, inconsistent with the study report by Jacobson, deGroot [18]. Females tended to show a lower QOL, which was like the finding in the Chinese population [40]. However, the presence of diabetes-related disease(s), marital status, and the gender of people living with diabetes revealed non-significant findings that were inconsistent with those from a previous study conducted in Botswana [39]. According to the previous studies, married participants had better QOL than those who were separated/widowed [18,24,39], which is consistent with the finding of this study. This may be because married participants might receive support from their families/partners, which might in turn boost their quality of life [24,41]. This finding demonstrated a significant worry due to diabetes. This finding was consistent with a study conducted in Spain [42].

The strengths of this study include the involvement of a large sample size to test the psychometric properties of the DQOL and the fact that factor-solutions were formed by EFA. The limitations of this study include the fact that the recruitment of the subjects was by convenience and thus may not be representative of all adults living with T2D who speak Afaan Oromoo in Ethiopia. However, the demographic characteristics of the participants in terms of gender, educational, and employment status are almost identical. In addition, there was a lot of missing data, possibly due to the length of the scale. In our study, we produced a shorter version of the DQOL for use with people with Type 2 diabetes who speak Afaan Oromoo, and this could have addressed the missing data issue associated with the original scale. Nevertheless, the hospital is one of the largest hospitals in Western Ethiopia, and many people living with diabetes attend medical check-ups there. Another limitation of this study was that test–retest reliability was not examined due to the COVID-19 pandemic, and the medical follow-up visits for diabetes in the center were delayed beyond the subject recruitment period.

## 5. Conclusions

The 34-item DQOL-AO provided preliminary evidence as a reliable and valid tool to measure QOL among adults living with T2D who speak Afaan Oromoo. Future research should assess the psychometric properties, like test-retest reliability, and the predictive validity of the 34-item DQOL-AO before it can be widely used among adults living with Type 2 diabetes who speak Afaan Oromoo.

## Figures and Tables

**Table 1 ijerph-18-07435-t001:** Sociodemographic characteristics of people living with type 2 diabetes attending Nekemte Specialized Hospital, 2020 (*n* = 417).

Variables with Categories	Frequency (%)
Age in years	Mean 50.2 (SD ± 11.7)
Gender	
Female	214 (51.3%)
Male	203 (48.7%)
Marital status	
Married	323 (77.5%)
Never married	30 (7.2%)
Separated/widowed	64 (15.3%)
Ethnicity	
Oromoo	368 (88.2%)
Amhara	43 (10.3%)
Other	6 (1.4%)
Religion	
Protestant Christian	237 (56.8%)
Orthodox Christian	138 (33.1%)
Muslim	34 (8.2%)
Other	8 (1.9%)
Primary caregiver (support provider)	
Spouse	257 (61.6%)
Children	123 (29.5%)
Mother or father	37 (8.9%)
Educational status	
No formal education	76 (18.2%)
Elementary school (≤ grade 8)	138 (33.1%)
Secondary school (grade 9–12)	101 (24.2%)
College and above	102 (24.5%)
Employment status	
Government employee	76 (18.2%)
Private organization employee	113 (27.2%)
Unemployed	73 (17.5%)
Retired/disabled	86 (20.6%)
Farmer	69 (16.5%)
Presence of disease-related disease	
Yes	231 (55.4%)
No	186 (44.6%)
Type of disease-related disease	
Hypertension	190 (82.3%)
Other diseases	41 (17.7%)
Years since diagnosis of diabetes	
≤10	345 (82.7%)
>10	72 (17.3%)

**Table 2 ijerph-18-07435-t002:** Item–total statistics of the DQOL-AO version among adults living with type 2 diabetes attending Nekemte Specialized Hospital, 2020 (*n* = 417).

Item Number	Domain and Items	Item–Total Correlation
Satisfaction		
1	How satisfied are you with the amount of time it takes to manage your diabetes?	0.581
2	How satisfied are you with the amount of time you spend getting check-ups?	0.543
3	How satisfied are you with the time it takes to determine your sugar level?	0.352
4	How satisfied are you with your current treatment?	0.461
5	How satisfied are you with the flexibility you have in your diet?	0.326
6	How satisfied are you with the burden your diabetes is placing on your family?	0.365
7	How satisfied are you with your knowledge about your diabetes?	**0.207**
8	How satisfied are you with your sleep?	0.351
9	How satisfied are you with your social relationships and friendships?	0.626
10	How satisfied are you with your sex life?	0.481
11	How satisfied are you with your work, school, and household activities?	0.543
12	How satisfied are you with the appearance of your body?	0.596
13	How satisfied are you with the time you spend exercising?	0.537
14	How satisfied are you with your leisure time?	0.496
15	How satisfied are you with life in general?	**0.223**
Impact		
1	How often do you feel the pain associated with the treatment of your diabetes?	0.363
2	How often are you embarrassed by having to deal with your diabetes in public?	0.303
3	How often do you have low blood sugar?	**0.181**
4	How often do you feel physically ill?	0.473
5	How often does your diabetes interfere with your family life?	0.518
6	How often do you have a bad night’s sleep?	0.412
7	How often do you find your diabetes limiting your social relationships and friendships?	0.486
8	How often do you feel good about yourself?	**−0.151**
9	How often do you feel restricted by your diet?	0.339
10	How often does your diabetes interfere with your sex life?	0.523
11	How often does your diabetes keep you from driving a car or using a machine (e.g., a typewriter)?	0.522
12	How often does your diabetes interfere with your exercising?	0.468
13	How often do you miss work, school, or household duties because of your diabetes?	0.554
14	How often do you find yourself explaining what it means to have diabetes?	0.359
15	How often do you find that your diabetes interrupts your leisure-time activities?	0.557
16	How often do you tell others about your diabetes?	**0.284**
17	How often are you teased because you have diabetes?	**0.245**
18	How often do you feel that because of your diabetes you go to the bathroom more than others?	**0.273**
19	How often do you find that you eat something you shouldn’t rather than tell someone that you have diabetes?	**−0.291**
20	How often do you hide from others the fact that you are having an insulin reaction?	**0.167**
Social/Vocational Worry		
1	How often do you worry about whether you will get married?	0.387
2	How often do you worry about whether you will have children?	0.429
3	How often do you worry about whether you will not get a job you want?	0.433
4	How often do you worry about whether you will be denied insurance?	**0.019**
5	How often do you worry about whether you will be able to complete your education?	**0.292**
6	How often do you worry about whether you will miss work?	0.404
7	How often do you worry about whether you will be able to take a vacation or a trip?	0.358
Diabetes-Related Worry		
1	How often do you worry about whether you will pass out?	0.496
2	How often do you worry that your body looks different because you have diabetes?	0.574
3	How often do you worry that you will get complications from your diabetes?	0.407
4	How often do you worry about whether someone will not go out with you because you have diabetes?	**0.116**

**Table 3 ijerph-18-07435-t003:** Initial EFA results of the DQOL-AO version (in English) among adults living with type 2 diabetes attending Nekemte Specialized Hospital, 2020 (*n* = 417).

Item Number	Item	Factor Loading			
		Factor 1	Factor 2	Factor 3	Factor 4
Satisfaction					
1	How satisfied are you with the amount of time it takes to manage your diabetes?		−0.742	0.503	
2	How satisfied are you with the amount of time you spend getting check-ups?		−0.705	0.569	
3	How satisfied are you with the time it takes to determine your sugar level?		−0.342	0.510	
4	How satisfied are you with your current treatment?		−0.536		
5	How satisfied are you with the flexibility you have in your diet?		−0.352		0.321
6	How satisfied are you with the burden your diabetes is placing on your family?	0.374			
7	How satisfied are you with your sleep?		−0.546		
8	How satisfied are you with your social relationships and friendships?	0.423	−0.448		
9	How satisfied are you with your sex life?		−0.497		
10	How satisfied are you with your work, school, and household activities?	0.726			
11	How satisfied are you with the appearance of your body?	0.567			
12	How satisfied are you with the time you spend exercising?	0.611			
13	How satisfied are you with your leisure time?	0.500			
Impact					
1	How often do you feel the pain associated with the treatment of your diabetes?	0.453			
2	How often are you embarrassed by having to deal with your diabetes in public?			0.391	
3	How often do you feel physically ill?	0.600			
4	How often does your diabetes interfere with your family life?	0.682			
5	How often do you have a bad night’s sleep?		−0.560		
6	How often do you find your diabetes limiting your social relationships and friendships?	0.431			
7	How often do you feel restricted by your diet?	0.401			
8	How often does your diabetes interfere with your sex life?	0.502	−0.308		
9	How often does your diabetes keep you from driving a car or using a machine (e.g., a typewriter)?	0.775			
10	How often does your diabetes interfere with your exercising?	0.709			
11	How often do you miss work, school, or household duties because of your diabetes?	0.614			
12	How often do you find yourself explaining what it means to have diabetes?	0.478			
13	How often do you find that your diabetes interrupts your leisure-time activities?	0.555		0.359	
Social/Vocational Worry					
1	How often do you worry about whether you will get married?		0.587	0.306	
2	How often do you worry about whether you will have children?		0.475	0.500	
3	How often do you worry about whether you will not get a job you want?			0.760	
4	How often do you worry about whether you will miss work?			0.673	
5	How often do you worry about whether you will be able to take a vacation or a trip?			0.389	
Diabetes-Related Worry					
1	How often do you worry about whether you will pass out?				0.765
2	How often do you worry that your body looks different because you have diabetes?				0.745
3	How often do you worry that you will get complications from your diabetes?				0.725

**Table 4 ijerph-18-07435-t004:** The diabetes quality of life-Afaan Oromoo version (in Afaan Oromoo) among people living with type 2 diabetes attending Nekemte Specialized Hospital, 2020 (*n* = 417).

**Qajeelfama: Maaloo tokkoon tokkoon hima armaan gadii sirriitti dubbisaa. Hagam akka haala jireenya keessan yeroo ammaa kan gaaffilee keessatti ibsaman itti quuftan ykn hin quufiin agarsiisaa. Lakkoofsa sirriitti miira keessan ibsu itti maraa. Nuti yaada keessan qofa baruu barbaanneeti malee gaaffilee kanaaf deebiin sirrii ykn dogoggoraa hin jiru.**							
**Koodii**	**Gaaffiilee ijoo**	**Deebii**					
	**Itti quufiinsa**	Baay’ee itti quufeera	Giddu-galeessaan itti quufeera	Giddu galeessa	Giddugaleessaan itti hin quufne	Baay’een itti hin quufnee	
1	Dhukkuba sukkaaraa keessan *yaalamuuf* yeroo isinitti fudhatu ilaalchisee hagam itti quufiinsa qabdu?	1	2	3	4	5	
2	Dhukkuba sukkaaraa keessan *ilaalamuuf* yeroo isinitti fudhatu ilaalchisee hagam itti quufiinsa qabdu?	1	2	3	4	5	
3	Hanga sukkaara keessan baruuf yeroo isinitti fudhatu ilaalchisee hagam itti quufiinsa qabdu?	1	2	3	4	5	
4	Yaalii amma isiniif godhamaa jirutti hagam quufiinsa qabdu?	1	2	3	4	5	
5	Jijjiirama haala soorata keessan keessatti qabdanitti hagam quufiinsa qabdu?	1	2	3	4	5	
6	Ba’aa/dhiibbaa dhukkubni sukkaaraa keessan maatii keessan irraan ga’aa jirutti hagam quufiinsa qabdu?	1	2	3	4	5	
7	Haala hirriba keessanitti hagam quufiinsa qabdu?	1	2	3	4	5	
8	Walitti-dhufeenya hawaasummaa fi hiriyummaa qabdanitti hagam quufiinsa qabdu?	1	2	3	4	5	
9	Haala wal-quunnamtii saalaa qabdanitti hagam quufiinsa qabdu?	1	2	3	4	5	
10	Gochaalee bakka hojiitti, mana-barumsaatti, akkasumas mana keessan keessatti raawwattanitti hagam quufiinsa qabdu?	1	2	3	4	5	
11	Dhaabbii/bifa qaama keessanitti hagam quufiinsa qabdu?	1	2	3	4	5	
12	Sochii ga’uumsa qaamaa gochuuf yeroo fudhattanitti hagam quufiinsa qabdu?	1	2	3	4	5	
13	Yeroo bashannanaa qabdanitti hagam quufiinsa qabdu?	1	2	3	4	5	
**Qajeelfama: Maaloo wantootni armaan gadii hagam deddeebi’anii akka isin mudatan argisiisaa. Lakkoofsa sirriidha jettanitti maraa.**							
**Gaaffiilee ijoo**		**Deebii**					
**Dhiibbaa dhukkubichi isin irraan gahu**		Tasa nah in mudanne	Baay’ee turee	Darbee darbee	Deddeebi’ee	Yeroo mara	
1	Dhukkubbiin (miidhaan) yaalii dhukkuba sukkaaraa keessan waliin walqabatu hagam deddeebi’ee isinitti dhaga’ama?	1	2	3	4	5	
2	Waa’ee dhibee sukkaaraa keessaniin wal-qabateen hawaasa gidduutti hagam deddeebitanii qaanoftanii/yeelloftanii/beektu?	1	2	3	4	5	
3	Dhukkubbiin qaamaa hagam deddeebi’ee isinitti dhaga’ama?	1	2	3	4	5	
4	Dhukkubni sukkaaraa keessan jireenya maatii keessan keessa hagam deddeebi’ee isin duraa seena/jeeqa?	1	2	3	4	5	
5	Hirriba halkanii badaa hagam deddeebitanii qabaattu?	1	2	3	4	5	
6	Dhukkubni sukkaaraa keessan walitti-dhufeenya hawaasummaafi hiriyummaa keessan hagam deddeebi’ee utuu daangessuu argitu?	1	2	3	4	5	
7	Haala soorannaa keessaniin daangeffamuun hagam deddeebi’ee isinitti dhaga’ama?	1	2	3	4	5	
8	Dhukkubni sukkaaraa keessan haala quunnamtii saalaa keessan keessa hagam deddeebi’ee isin duraa seena?	1	2	3	4	5	
9	Dhukkubni sukkaaraa keessan konkolaataa oofuurraa ykn maashinii fayyadamuu (fkn maashinii barreeffamaa) ykn hojii guyyaa guyyaan hojjattan irraa hagam deddeebi’ee isin dhorka?	1	2	3	4	5	
10	Dhukkubni sukkaaraa keessan sochii ga’uumsa qaamaa isin gootan keessa hagam deddeebi’ee isin duraa seena?	1	2	3	4	5	
11	Sababa dhukkuba sukkaaraa keessaniin bakka hojii, mana barumsaa ykn hojii mana keessaa hagam deddeebitanii irraa haftu?	1	2	3	4	5	
12	Dhukkuba sukkaaraa qabaachuu jechuun maal akka ta’e utuu ibsitanii hagam deddeebitanii of agartu?	1	2	3	4	5	
13	Dhukkubni sukkaaraa keessan kun yeroo bashannanaa keessan utuu addaan-kutuu hagam deddeebitanii argitu?	1	2	3	4	5	
**Qajeelfama: Maaloo taateewwan armaan gadii hagam deddeebi’anii akka isin quunnaman argisiisaa. Lakkoofsa sirriitti miira keessan isiniif ibsutti maraa. Gaaffichi isin hin ilaallatu yoo ta’emmoo, ‘na hin ilaallatu’ kan jedhutti maraa.**							
	Gaaffiilee ijoo waa’ee yaaddoo hawaasummaa/hojii ogummaa	Tasuma	Baay’ee turee	Darbee darbee	Deddeebi’ee	Yeroo mara	Na hin ilaallatu
1	Gara fuulduraatti waa’ee gaa’ela dhaabbachuu keessan hagam deddeebitanii yaaddoftu?	1	2	3	4	5	0
2	Gara fuulduraatti waa’ee ijoollee argachuu keessan hagam deddeebitanii yaaddoftu?	1	2	3	4	5	0
3	Gara fuulduratti dhibee kana irraan kan ka’e waa’ee hojii barbaaddan argachuu dhabuu keessan hagam deddeebitanii yaaddoftu?	1	2	3	4	5	0
4	Hojiikoo irraan hafaa laata jettanii hagam deddeebitanii yaaddoftu?	1	2	3	4	5	0
5	Boqonnaaf ykn bashannanaaf bakka biraa deemuu danda’uu keessan hagam deddeebitanii yaaddoftu?	1	2	3	4	5	0
**Dubbii ijoo yaaddoo dhukkuba-sukkaaraan wal qabatan Deebii**							
1	Dhukkubni kun na of-wallaalchisaa laata jettanii hagam deddeebitanii yaadoftu?	1	2	3	4	5	0
2	Sababa dhukkuba sukkaaraa kanaaf qaamnikoo ni jijjiirame jettanii hagam deddeebitanii yaaddoftu?	1	2	3	4	5	0
3	Sababa dhukkuba-sukkaaraa keessaniin kan ka’e dhibeen walxaxaan natti dhufa jettanii hagam deddeebitanii yaaddoftu?	1	2	3	4	5	0

**Table 5 ijerph-18-07435-t005:** The internal consistency, ceiling, and floor effects of the DQOL-AO version among people living with type 2 diabetes attending Nekemte Specialized Hospital, 2020 (*n* = 417).

Scale/Subscale	Number of Items	Cronbach’s Alpha	Mean Score (±SD)	Ceiling Effect *n* (%)	Floor Effect *n* (%)
Impact	13	0.827	2.43 (0.49)	2 (0.5)	1 (0.2)
Satisfaction	13	0.846	2.46 (0.56)	1 (0.2)	1 (0.2)
Social/vocational worry	5	0.654	1.53 (0.93)	26 (6.2)	2 (0.5)
Diabetes-related worry	3	0.727	3.20 (0.65)	2 (0.5)	4 (1.0)
Total DQOL	34	0.867	2.38 (0.43)	1 (0.2)	1 (0.2)

**Table 6 ijerph-18-07435-t006:** Relationships between DQOL-AO scores and demographic variables and pairwise comparison among people living with type 2 diabetes attending Nekemte Specialized Hospital, 2020 (*n* = 417).

Variables with Categories	Frequency (n)	Mean Score	SD	Between-Group *p*-Value	Pairwise Comparison
Gender					
Female	214	2.40	0.42	0.288	
Male	203	**2.36**	0.42		
Age	217	r = 0.057		0.247	
Diabetes-specific complication(s) status					
No	186	**2.35**	0.43	0.163	
Yes	231	2.41	0.42		
Marital status					
Married	323	**2.36**	0.42	0.352	
Never married	30	2.43	0.47		
Separated/widowed	64	2.44	0.43		
Educational status					
No formal education	76	2.43	0.43	<0.001	College and above < elementary school, no formal education
Elementary school (≤ grade 8)	138	2.47	0.42		
Secondary school (grade 9–12)	101	2.37	0.42		
College and above	102	**2.23**	0.40		
Employment status					
Government employees	76	**2.23**	0.39	0.02	Government employees < retired/disabled, farmer
Private organization employees	113	2.35	0.45		
Unemployed	73	2.39	0.41		
Retired/disabled	86	2.44	0.39		
Farmer	69	2.39	0.44		

r = Pearson correlation coefficient.

## Data Availability

The dataset is not available publicly but is available from Dereje Chala Diriba on request.

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
