# Peer review of "Cultural Adaptation and Psychometric Properties of the Diabetes Quality of Life Scale in Afaan Oromoo among People Living with Type 2 Diabetes in Ethiopia"

_ijerph, 2021, doi:10.3390/ijerph18147435_

Round 1
Reviewer 1 Report
Thank you for the opportunity to review this paper. There is a clear rationale for the study, addressing a potential health inequality, and the authors have provided good attention to detail in terms of transparency about how the translated scale was developed. Before its publication, however, I believe there are some issues that need addressing. Most of these relate to the need for clarification or improved writing. Please see below for specific comments.
Abstract:
l. 9 -10 This is labelled background but contains no background. Rather these lines describe the aims.
L.20 ‘in’ should be replaced with ‘between’.
L.23 ‘As’ should be replaced with ‘for’.
Introduction
L.31 Consider replacing the word ‘rampant’. It’s a bit dramatic for academic writing.
L.33 What does the 13% refer to?
L.33 Increment of what?
L.38-39 Unclear sentence.
L.43 ‘complications’. Does this refer to ‘complications of diabetes’? It could be clearer if so.
L.48 ‘Compared to…’ This sentence needs clarification and/or expansion. It currently doesn’t really provide any helpful information.
L.47 onwards: Generally this paragraph is a bit repetitive and could be re-written more concisely.
L.69-70 Not clear if these values are referring to all the aforementioned studies, or the original scale development.
L.73 onwards: Can the authors spell out specifically why this relates to construct validity?
Methods
L.96 It is conventional to include the Participants demographics table round about here in the Methods rather than in the Results.
L.123 It would be helpful to expand somewhere (probably in the Results) as to whether the translation process created any issues. It seems the process was very thorough so it would be good to know if these reliability checks highlighted any problem items or problem translations.
L.128 Why EFA, not CFA? The latter is more conventional when conducting scale validation of an already existing scale. I think EFA can be justified but it would be helpful to expand on the analysis choice here.
L.129 10:1 ratio: There is a lot of debate around this ration. The rule of thumb is out of date and therefore requires more justification than this. At the very least, some references to support the ratio would improve this section. Ideally, a better reason for the sample size than ‘rule of thumb’.
L.130 Included? Or required?
L.175 This is very unclear. I would think 9.4% is quite high. And what about the others? Or perhaps I have misunderstood, in which case the sentence need clarifying.
Results
L.177 This looks more like the authors are saying that 9.4% of participants didn’t complete all the questions? If this is true, state this as this is not what it sounds like in the previous sentence. Overall these few sentences need clarification.
L.181 Onwards: The paragraph is quite repetitive of the table. Consider removing some of the text. As above, I also think that table belongs in a Participants section in the Methods.
L.192 This is a lot of items. Do we need to consider whether the scale is more widely not appropriate for this population (Or justify why it still is)? Perhaps something for the Discussion.
Table 2: Formatting issues.
L.219 It would be helpful to signpost the reader to the Table including the alpha values here.
L.232 onwards. There needs to be an indication for each difference what the direction of the difference was. Or direct people to the Table and drop the text. As it stands the paragraph does not stand alone with helpful information.
Table 5: I assume the bold numbers are to indicate where the differences lie. This speaks for itself with the tests that have more than two groups but is a bit confusing for the first few variables. IT would be helpful to state somewhere that this is what the bold represents (or whatever it does represent if I’m wrong).
Discussion:
L.257 The authors are very detailed in their description of what was dropped and why. However, it is difficult to read and follow without having to constantly reference what item is which from the earlier tables. I suggest re-writing this information to make broader conclusions about why certain items didn’t fit. I think some of the information belongs better in the Results (e.g. specific reasons why items were dropped like the insulin related ones which are less relevant for this population). The paragraph currently reads like a list, where it would be more appropriate in the Discussion to synthesise the information and think about the bigger picture: What are the implications of the dropped items? Can you still be confident in the scale? Is there any appropriate theory to draw in? Etc.
L.282 -284 Can you expand?
L.300 Consider why this is different in this sample to similar studies elsewhere.
L.301-2 Please rephrase, this sentence doesn’t currently make any sense.
L.307 It would be helpful to add a comment about how representative this sample is of the population (i.e. consider the demographics and proportions).
L.335 Is there a reason not to make the scale available publicly? It seems from all the analysis that it would be technically possible to work it out anyway? And it might be neater and more helpful to include it and present how the paper should be references if people choose to use it?
Author Response
Dear Editor,
The point-to-point responses to the reviewer's comment are uploaded in word doc. file. Please see the attachment.

Reviewer 2 Report
The paper is a cultural adaptation of the QOL scale to the Afaan Oromoo. The paper has a clinical relevance because it allowed clinician to use a validated questionnaire, even if the impact of the paper could be very low. I think the sample is sufficient for the scope of the paper. I have only few suggestion for the authors: - I think that the factorial analysis should be done without the use of the original subscales because after translation factors could be different than the original ones. - I think the discussion could be partially revised because it is more a justification of the results. Beside these comments, I think that the paper is in line with the goal of the journal.Author Response
Dear Editor,
The point-to-point responses to the reviewer's comments are uploaded separately.
Please see the attachment.
Regards,
Dereje Chala Diriba (on behalf of the authors)
